# Presyncodon, a Web Server for Gene Design with the Evolutionary Information of the Expression Hosts

**DOI:** 10.3390/ijms19123872

**Published:** 2018-12-04

**Authors:** Jian Tian, Qingbin Li, Xiaoyu Chu, Ningfeng Wu

**Affiliations:** 1Biotechnology Research Institute, Chinese Academy of Agricultural sciences, Beijing 100081, China; tianjian@caas.cn (J.T.); liqingbin2015@sina.cn (Q.L.); chuxiaoyu@caas.cn (X.C.); 2State Key Laboratory of Agrobiotechnology, College of Biological Sciences, China Agricultural University, Beijing 100081, China

**Keywords:** gene design, presyncodon, expression host, codon optimization, web server

## Abstract

In the natural host, most of the synonymous codons of a gene have been evolutionarily selected and related to protein expression and function. However, for the design of a new gene, most of the existing codon optimization tools select the high-frequency-usage codons and neglect the contribution of the low-frequency-usage codons (rare codons) to the expression of the target gene in the host. In this study, we developed the method Presyncodon, available in a web version, to predict the gene code from a protein sequence, using built-in evolutionary information on a specific expression host. The synonymous codon-usage pattern of a peptide was studied from three genomic datasets (*Escherichia coli*, *Bacillus subtilis*, and *Saccharomyces cerevisiae*). Machine-learning models were constructed to predict a selection of synonymous codons (low- or high-frequency-usage codon) in a gene. This method could be easily and efficiently used to design new genes from protein sequences for optimal expression in three expression hosts (*E. coli*, *B. subtilis*, and *S. cerevisiae*). Presyncodon is free to academic and noncommercial users; accessible at http://www.mobioinfor.cn/presyncodon_www/index.html.

## 1. Introduction

In most organisms, 61 universal genetic codons encode for 20 standard amino acids, of which 18 are encoded by multiple synonymous codons. In all domains of life, a biased frequency of synonymous codons is observed at the genome level. Many studies have proved that the presence of synonymous codons in the gene coding regions is not inconsequential, and relates to the efficient and accurate translation of the protein [1,2,3]. Therefore, codon optimization can affect protein expression and function in the heterologous gene expression system [4,5,6,7].

Many methods, including JCat [8], Gene Designer [9], OPTIMIZER [10], Gene Composer [11], COStar [12], and COOL [13] have been proposed to design heterologous genes that are expected to be efficiently expressed in the host organism. Based on our experience, we concluded that these methods are prone to select the high-frequency-usage codons and neglect the contribution of the low-frequency-usage codons (rare codons) to the expression of the target gene. However, in the case of some genes, single point synonymous codons can also affect the expression and function of the target protein [4,14,15,16]; and some rare codons are conserved in the evolution and play an important role to regulate protein folding and protein production [16,17,18]. Therefore, those methods have an over-reliance on the prediction and usage of codons that are frequently selected in highly-expressed genes. In the natural host, most of the synonymous codons of the gene have been evolutionarily selected and; therefore, in order to account for all the evolutionary variation, the codon usage pattern should be learned from the natural genes [16,19].

To address the need for heterologous gene design, based on all used codons (the high- or low-frequency-usage codons), a new web server application, Presyncodon, was developed to design the heterologous gene for expression in the three frequently-used recombinant hosts (*Escherichia coli*, *Bacillus subtilis*, and *Saccharomyces cerevisiae*). This big data gene prediction method was used to learn the codon-usage pattern for a peptide as-derived from sequenced genomic data. Machine-learning models were constructed by the random forest classification to predict a selection of synonymous codons (low- or high-frequency-usage codon) for the target gene. Compared with the early version of Pyesyncodon, which could only design the gene to be efficiently expressed in *E. coli* in local [20], this new version could design new genes from protein sequences for optimal expression in three recombinant hosts (*E. coli*, *B. subtilis*, and *S. cerevisiae*) on the web; and the training dataset has been updated with more genomes. Therefore, this method will be easily and efficiently used to design genes for heterologous gene expression in the three popular expression hosts (*E. coli*, *B. subtilis*, and *S. cerevisiae*).

## 2. Materials and Methods

### 2.1. Dataset

Three genomic datasets (*E. coli*, *B. subtilis*, and *S. cerevisiae*) were constructed, which contained 353, 62, and 20 genomes, respectively. All selected genomes were the complete genomes downloaded from NCBI (ftp://ftp.ncbi.nlm.nih.gov/genomes/) on July 13th, 2016, and their genome accession numbers are shown in Appendix A.

In order to train the predicting models, three non-redundant gene datasets (*E. coli*, *B. subtilis*, and *S. cerevisiae*) were also constructed. The software CD-HIT [21] was used to calculate the gene clusters and remove the redundant genes in the cluster with protein sequences exhibiting over 40% identity. For the aim to remove the peculiar genes that might evolve from the horizontal transfer, the typical genes from those gene clusters that contained at least three homologous sequences were selected. The required length of each sequence was over 100 codons. As a result, three gene datasets, covering *B. subtilis*, *E. coli* and *S. cerevisiae*, were constructed with 8091, 11232, and 5905 genes from the total 1461067, 256246, and 107820 genes, respectively. 

### 2.2. Workflow

The general flowchart of the method is shown in Figure 1. Firstly, each gene in the constructed gene database was split into window sizes of five and seven codons. Then a codon selection index (CSI) for each set of genomic data (five and seven residues) was determined, which represented the codon usage distribution for the middle amino acid and the average codon usage for each amino acid in the fragment. 

The training gene sequences were translated, and were also split into window sizes of five or seven amino acids, and searched against the corresponding CSI files. For each fragment, the matched score (*s*), expected maximal score (*m*) of the target fragment, and the matched percent (*p*, *p* = *s*/*m*) against the CSI file were calculated by the method described in [20]. For a given cut-off level (*c*), if the calculated matched percent of multiple fragments from the CSI file for a fragment was greater than the cut-off level, the coding vector for the middle codon in the fragment was the arithmetic average of those vectors encoding the selected multiple fragments. Here, the training label is the codon for the middle amino acid.

All training labels and input vectors were collected, and the random forest classifier from “R” statistics package (ver.3.4.0) [22,23] was used to train the predicting models with seven cut-off levels (0.7, 0.75, 0.8, 0.85, 0.9, 0.95, and 1), two window sizes (5 and 7 residues), and 18 amino acids (containing multiple synonymous codons). The dimensionality of the input features for each amino acid was the codon number of the amino acid plus the window size. The number of trees of the key parameter of the classifier for random forests was set to 10,000. For each organism, 252 models (252 = 7 × 2 × 18) were constructed.

In order to increase the speed of target gene design, all possible fragments (a total of 2,880,000 (18 × 20^4^) in case of the five residues’ long fragments and of 1152000000 (18 × 20^5^) in case of the seven residues long fragments) were searched against the corresponding CSI files, with four cut-off levels (0.7, 0.8, 0.9, and 1). Input vectors were generated for each fragment and the synonymous codons selection, based on the distribution of the middle residue in the fragment, was predicted for each organism. The results were stored in the PostgreSQL database.

As the training vector only encodes for the middle codon in the fragment, the first and last two codons of a gene could not be predicted by the above machine learning models. The codon usage pattern was generated by measuring the codon-usage bias of the first and last two residues. Therefore, the first and last two codons of a gene were designed as the most frequently used codons at these positions in all genes (Appendix A).

## 3. Validation

The performance of the predicting models, obtained from the two fragment window sizes (5 and 7 amino acids) and cut-off level (*c*: 0.7, 0.75, 0.8, 0.85, 0.9, 0.95 and 1), were evaluated by ten-fold cross validation. As shown in Figure 2, the predicting accuracy of models obtained from the window size of seven amino acids was higher than that of the models obtained from the window size of five amino acids. Additionally, the classifier obtained with the larger cut-off level (*c*) achieved higher accuracy than those obtained with smaller cut-off levels. Therefore, the codon-usage tendency for each amino acid could be predicted by only one model, as obtained from the long-window-sized amino acid fragments and characterized by a larger cut-off level (*c*). The first and last two codons were selected statistically (Appendix A).

## 4. Implementation

The software Presyncodon is designed as an adaptable, web-based interface that could be easily used by scientists. This website was built using Linux (Centos ver. 6.5), Apache (ver. 2.2), PostgreSQL (ver. 8.4.20), and Perl (ver. 5.10.1). The input of the user is the target protein sequence and the only external parameter required is the selection of the target expression host (Figure 3). The waiting time for optimizing a 100-amino acid sequence is estimated to be two minutes. Therefore, the method could be easily used to design synthetic genes for heterologous gene expression in biotechnology. Based on this method, we have successfully designed the genes of GFP [20], mApple [20], laccase, penicillin-binding protein, alpha-1,4 glucan phosphorylase L-1 isozyme, pirin-like protein, and cadmium-binding proteins from maize to be efficiently expressed in *E. coli*. Now, this version of Presyncodon could be used to design the heterologous genes for expression in the three frequently-used recombinant hosts (*E. coli*, *B. subtilis*, and *S. cerevisiae*). In the next step, we will develop this optimizing method for more expression systems. Therefore, this method could be easily used to design synthetic genes for heterologous gene expression in biotechnology.

## Figures and Tables

**Figure 1 ijms-19-03872-f001:**
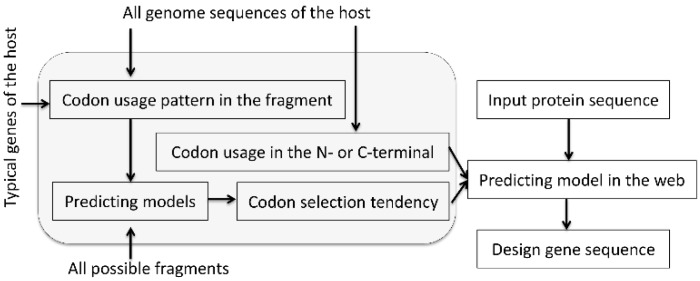
Flowchart summarizing the Presyncodon approach.

**Figure 2 ijms-19-03872-f002:**
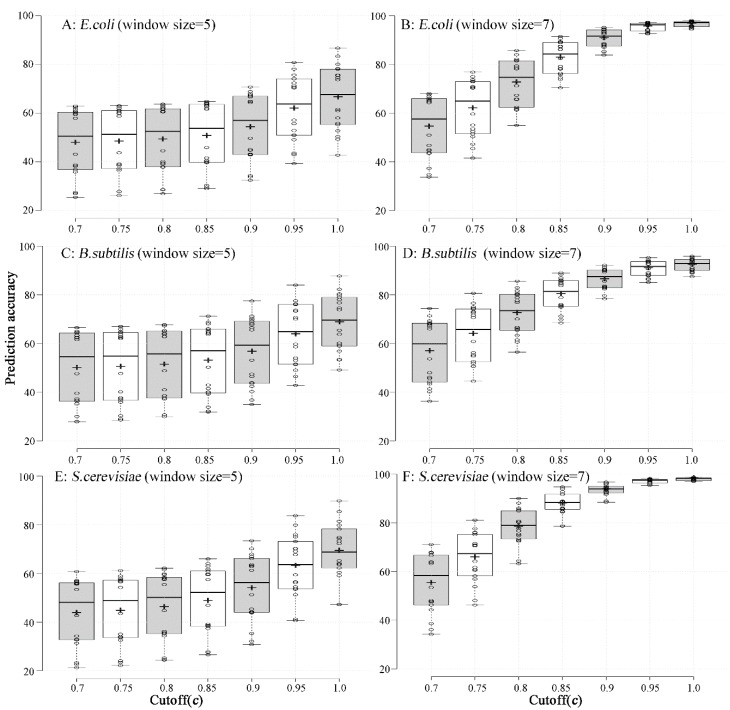
The prediction performance of the 18 classifiers for the 18 amino acids, with different matched cut-offs and window sizes (left: Five amino acids; right: Seven amino acids) in *E. coli*, *B. subtilis*, and *S. cerevisiae*. The *x*-axis is the matched percent and the *y*-axis is the prediction accuracy of the 18 classifiers. Each open circle represents the prediction accuracy with one of the 18 classifiers. The horizontal divisions (from top to bottom) in each box are the upper whisker, 3rd quartile, median, 1st quartile, and lower whisker, respectively. The cross line in each box is the mean prediction accuracy of all 18 classifiers. All of the results were calculated based on a ten-fold cross validation.

**Figure 3 ijms-19-03872-f003:**
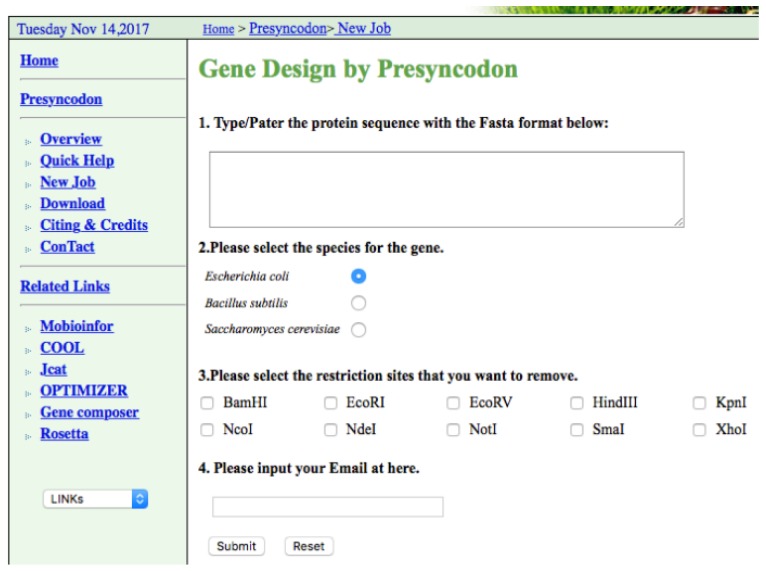
Screenshot of the web version of Presyncodon.

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
