# Peer review of "Presyncodon, a Web Server for Gene Design with the Evolutionary Information of the Expression Hosts"

_ijms, 2018, doi:10.3390/ijms19123872_

Round 1
Reviewer 1 Report
The authors are using web-based application to predict novel genes by using existing protein sequences. The gene model was developed based upon three available expression system. However, there are somethings that should be clarified:
1. Your expression system is too limited because it could only predict three expression system.
In the future venue of this work, I suggest to develop for these system as well:
Algae
Bacterial excl. E.coli
Cell free
Fungi
Insect Cells
Mamalian Cells
Yeast Cells
2. In the output of your application, you should provide the information of other possibility of predicted genes, as well as their probability.
3. What is your criteria to select 353, 62, and 20 genomes, respectively, for the downloads? Did you download all the genomes in the repository?
4. You said that genes for GFP, mApple, laccase, Penicillin-Binding Protein, Alpha-1,4 glucan phosphorylase L-1 120 isozyme, Pirin-like protein and cadmium binding proteins have been designed. You should deposit your design in online repository such us Zenodo so the reader could validate your claim.
5. You should mention the version number of these following software in your manuscript: R statistics package, Linux (and its distribution), Apache, PostgreSQL, and Perl
Author Response
Response to Reviewer 1 Comments
The authors are using web-based application to predict novel genes by using existing protein sequences. The gene model was developed based upon three available expression system. However, there are somethings that should be clarified:
1. Your expression system is too limited because it could only predict three expression system.
In the future venue of this work, I suggest to develop for these systems as well:
Algae
Bacterial excl. E.coli
Cell free
Fungi
Insect Cells
Mamalian Cells
Yeast Cells
Response: Thanks for your comments. In the future works, we will develop the optimizing method for those expression systems.
2. In the output of your application, you should provide the information of other possibility of predicted genes, as well as their probability.
Response: At the end of the prediction results, we provided the statistic or Machine learning models that were used to design the target genes.
3. What is your criteria to select 353, 62, and 20 genomes, respectively, for the downloads? Did you download all the genomes in the repository?
Response: Those genomes were the complete genomes and fully annotated in the NCBI. Yes, we downloaded and analyzed those genomes by our methods. And their genome IDs were shown in Table S1.
4. You said that genes for GFP, mApple, laccase, Penicillin-Binding Protein, Alpha-1,4 glucan phosphorylase L-1 120 isozyme, Pirin-like protein and cadmium binding proteins have been designed. You should deposit your design in online repository such us Zenodo so the reader could validate your claim.
Response: The sequences of designed genes of egfp and mapple were shown in the published paper (Scientific reports, 2017, 9926.). We have added the reference for those proteins. The designed sequences of the other proteins were not published. Therefore, it is difficult to post those sequences on line.
5. You should mention the version number of these following software in your manuscript: R statistics package, Linux (and its distribution), Apache, PostgreSQL, and Perl
Response: Thanks for the comment. We have added the version of those software in the manuscript.
Reviewer 2 Report
The work by Tian et al. presents an interesting tool for the design of genes based on the amino acid sequence of a protein as the sole external input. This study presents an interesting approach to deduce the most likely codon composition of a gene based on the "learning" of common patterns in the target organism, in this way adding context to the prediction based on other genes from the same organism. This tool could also be used to gain more insights on the variations of codon usage bias between different organisms.
Nevertheless, there are some issues that need to be addressed in order to provide a sound product that can attract and be useful to readers. Following my suggestions to improve the present report:
Abstract: In this sentence "for the design of a new gene, most of the existing codon optimization tools select the high-frequency-usage codons and neglect the low-frequency-usage codons of the host." It would be useful to add the reason for considering low frequency usage codons; it sticks to reason that "rare" codons are in less abundance in the translation pools of tRNA, then, why using those "rare" codons should be important in the tools?
Introduction: This sentence "In nature, 61 possible codons encode for 20 common amino acids, of which 18 are encoded by multiple synonymous codons." needs revision, as this description corresponds to the universal genetic code and some organisms and organelles present exceptions and changes (e.g. mitochondrial DNA).
Introduction line 29: delete "including Many methods,", as it seems a duplication of text.
Introduction line 31: In this sentence "Based on our experience, we concluded that these methods are prone to select the high-frequency-usage codons and neglect the contribution of the low-frequency-usage codons (rare codons) to the expression of the target gene." it could be possible to provide more evidence than anecdotal perception. For example, the contrast with the previous version of Presyncodon, this could be a good place to introduce the limitations of that version. Also, another possible reason could be listing the reasons behind the conservation of those "rare" codons despite the selection of other more efficient codons (possibly in higher abundance in the tRNA pools). Finally, it could be possible to emphasize that the flaw of other tools is over-reliance on the recognition and usage for prediction of codons frequently selected in highly expressed genes.
Introduction line 41: there is mention of Machine Learning, but there is not mention of what method was used or what type of Machine Learning approaches were considered for this study. The answer to this question is buried in the Materials and Methods section, this could be suspicious for some readers.
Introduction line 42: modify "for the target the gene" to "for the target gene".
Introduction lines 44-5: there is a mention of three recombinant hosts, however, up to this point there was no mention of recombinations or any other explanation about it.
Materials & Methods lines 58-9: this description "For the aim to find the conserved gene in the species, every gene should also have had at least three homologous sequences in the corresponding genomic dataset", why 3 sequences? Does this mean in the same genome having at least three homologs? Does this mean that in the group of genomes for the same species having multiple homologous? In any case, why 3?
Materials & Methods lines 60-1: In this sentence "with 8091, 11232 and 5905 genes, respectively." out of how many considered considered genes. Providing the information of the total can give a perspective about how strict was the filter and how much data was removed.
M&M lines 63-4: In this sentence "the gene sequences in the genomic database were split into window sizes of five or seven amino acids.", maybe it is better to clarify that it is "five and seven", as the results are done for both cases. According to my understanding, both partitions were done for each gene independently. If I am wrong, it would be useful to explain in what cases the window size was 5 codons and in what cases the size was 7 codons. Also, it is important to be clear with the use of "codon" and "amino acid", many descriptions are technically wrong as the genes should be described in codons or nucleotides and proteins should be described in amino acids. For example, the proper description of the sentence above should be: "the gene sequences in the genomic database were split into window sizes of five and seven codons."
M&M lines 73-5: this description "For a given cut-off level c, if the matched percent of multiple fragments from the CSI file was greater than the cut-off level, the coding vector for the target codon in the fragment is the arithmetic average of all of the matched record vectors." is not clear. Having all the fragments above the cut-off level, how the arithmetic average allows the tool to predict the nucleotide composition?
M&M lines 79-80: why 18 codons? What values were considered as correct predictions? I assume the original codon sequence from the genomes was used as the correct answer, however, it is not explained in the text.
M&M line 90: this sentence "As the first and last two codons of a gene could not be predicted by the above method", why they were not predictable by this method?
I just reviewed the webpage of the tool for some minutes, but I also recommend an English revision of the contents for better results with users and more frequent usage than other tools with similar goals.
Author Response
Response to Reviewer 2 Comments
The work by Tian et al. presents an interesting tool for the design of genes based on the amino acid sequence of a protein as the sole external input. This study presents an interesting approach to deduce the most likely codon composition of a gene based on the "learning" of common patterns in the target organism, in this way adding context to the prediction based on other genes from the same organism. This tool could also be used to gain more insights on the variations of codon usage bias between different organisms.
Nevertheless, there are some issues that need to be addressed in order to provide a sound product that can attract and be useful to readers. Following my suggestions to improve the present report:
Point 1. Abstract: In this sentence "for the design of a new gene, most of the existing codon optimization tools select the high-frequency-usage codons and neglect the low-frequency-usage codons of the host." It would be useful to add the reason for considering low frequency usage codons; it sticks to reason that "rare" codons are in less abundance in the translation pools of tRNA, then, why using those "rare" codons should be important in the tools?
Response: We have added a short sentence in the abstract to illustrate the contribution of the low-frequency-usage codons (rare codons) to the expression of the target gene in host.
Point 2. Introduction: This sentence "In nature, 61 possible codons encode for 20 common amino acids, of which 18 are encoded by multiple synonymous codons." needs revision, as this description corresponds to the universal genetic code and some organisms and organelles present exceptions and changes (e.g. mitochondrial DNA).
Response: Thanks for the comment. A modifier (Most of the organisms) was added to eliminate the exceptions.
Point 3. Introduction line 29: delete "including Many methods,", as it seems a duplication of text.
Response: Thanks. We have deleted those redundant words.
Point 4. Introduction line 31: In this sentence "Based on our experience, we concluded that these methods are prone to select the high-frequency-usage codons and neglect the contribution of the low-frequency-usage codons (rare codons) to the expression of the target gene." it could be possible to provide more evidence than anecdotal perception. For example, the contrast with the previous version of Presyncodon, this could be a good place to introduce the limitations of that version. Also, another possible reason could be listing the reasons behind the conservation of those "rare" codons despite the selection of other more efficient codons (possibly in higher abundance in the tRNA pools). Finally, it could be possible to emphasize that the flaw of other tools is over-reliance on the recognition and usage for prediction of codons frequently selected in highly expressed genes.
Response: Thanks for the comment. Two sentences have been added to introduce the function of rare codon and flaw of other tools.
Point 5. Introduction line 41: there is mention of Machine Learning, but there is not mention of what method was used or what type of Machine Learning approaches were considered for this study. The answer to this question is buried in the Materials and Methods section, this could be suspicious for some readers.
Response: The machine learning models were constructed by the classifier of random forest. We have supplied this information in the Materials and Methods section.
Point 6. Introduction line 42: modify "for the target the gene" to "for the target gene".
Response: Thanks. We have revised it.
Point 7. Introduction lines 44-5: there is a mention of three recombinant hosts, however, up to this point there was no mention of recombinations or any other explanation about it.
Response: We have revised the first sentence in this paragraph to introduce theses three frequently-used recombinants.
Point 8. Materials & Methods lines 58-9: this description "For the aim to find the conserved gene in the species, every gene should also have had at least three homologous sequences in the corresponding genomic dataset", why 3 sequences? Does this mean in the same genome having at least three homologs? Does this mean that in the group of genomes for the same species having multiple homologous? In any case, why 3?
Response: This requirement is just to remove the peculiar genes that might evolve from the horizontal transfer. We have revised those sentences to clear this aim, as the follows. The software CD-HIT was used to calculate the gene clusters and remove the redundant genes in the cluster with protein sequences exhibiting over 40% identity. For the aim to remove the peculiar genes that might evolve from the horizontal transfer, the typical genes from those gene clusters that contained at least three homologous sequences were selected.
Point 9. Materials & Methods lines 60-1: In this sentence "with 8091, 11232 and 5905 genes, respectively." out of how many considered genes. Providing the information of the total can give a perspective about how strict was the filter and how much data was removed.
Response: The total gene number in the gene dataset was supplied in the manuscript, as the follows. As a result, three gene datasets covering B. subtilis, E. coli and S. cerevisiae were constructed with 8091, 11232 and 5905 genes from the total 1461067, 256246 and 107820 genes, respectively.
Point 10. M&M lines 63-4: In this sentence "the gene sequences in the genomic database were split into window sizes of five or seven amino acids.", maybe it is better to clarify that it is "five and seven", as the results are done for both cases. According to my understanding, both partitions were done for each gene independently. If I am wrong, it would be useful to explain in what cases the window size was 5 codons and in what cases the size was 7 codons. Also, it is important to be clear with the use of "codon" and "amino acid", many descriptions are technically wrong as the genes should be described in codons or nucleotides and proteins should be described in amino acids. For example, the proper description of the sentence above should be: "the gene sequences in the genomic database were split into window sizes of five and seven codons."
Response: Thanks for the carful comment. We have revised it based on this comment. The second sentence in this paragraph was revised to “The each gene in the constructed gene database was split into window sizes of five and seven codons”.
Point 11. M&M lines 73-5: this description "For a given cut-off level c, if the matched percent of multiple fragments from the CSI file was greater than the cut-off level, the coding vector for the target codon in the fragment is the arithmetic average of all of the matched record vectors." is not clear. Having all the fragments above the cut-off level, how the arithmetic average allows the tool to predict the nucleotide composition?
Response: Thanks for the comment. We have revised it and cleared the concept. Only few fragments have the matched percent above the cut-off level.
Point 12. M&M lines 79-80: why 18 codons? What values were considered as correct predictions? I assume the original codon sequence from the genomes was used as the correct answer, however, it is not explained in the text.
Response: In lines 79-90, it is not 18 codons, but 18 amino acids. We only constructed the models for predicting the codon selection in the 18 amino acids, as there are only 18 amino acids containing the multiple synonymous codons.
Point 13. M&M line 90: this sentence "As the first and last two codons of a gene could not be predicted by the above method", why they were not predictable by this method?
Response: As the training vector only encodes for the middle codon in the fragment, the first and last two codons of a gene could not be predicted by the machine learning models. We have revised this sentence.
Point 14. I just reviewed the webpage of the tool for some minutes, but I also recommend an English revision of the contents for better results with users and more frequent usage than other tools with similar goals.
Response: Thanks for the comment and visiting our website. We are trying our best for the scientists to use this tool easily.
Round 2
Reviewer 1 Report
The authors have clarified the posed questions in good manner.